# Home intravenous diuretic administration for heart failure management: A scoping review

**Morgan B. Krauter**[1]*, **Katherine S. McGilton**[1,2], **Stuti S. Patel**[2], **Karen Harkness**[3], **Tracey J. F. Colella**[1,4]

1 Lawrence S. Bloomberg Faculty of Nursing, University of Toronto, Toronto, ON, Canada, 2 KITE Research Institute, Toronto Rehabilitation Institute, University Health Network, Toronto, ON, Canada, 3 McMaster University, Hamilton, Ontario, Canada, 4 Cardiovascular Prevention and Rehabilitation Program, Toronto Rehabilitation Institute, University Health Network, Toronto, ON, Canada

* m.krauter@mail.utoronto.ca

## Abstract

### Background

Heart failure (HF) significantly impacts healthcare systems due to high rates of hospital bed utilization and readmission rates. Chronic HF often leads to frequent hospitalizations due to recurrent exacerbations and a decline in patient health status. Intravenous (IV) diuretic administration is essential for treating worsening HF. Emerging strategies include home-based IV diuretic therapy administration; however, limited practical implementation guidance is available. This scoping review aims to summarize the literature on home IV diuretic administration for HF patients, focusing on the interventions' characteristics, and facilitators and barriers to its implementation.

### Methods

This review followed the scoping review framework proposed by Arksey and O'Malley and PRIMSA-ScR. A comprehensive search was conducted across six databases (CINAHL, the Cochrane Library, EMBASE, MEDLINE, PsychINFO and Scopus) and grey literature to identify English studies from inception to April 13, 2024. Two independent reviewers screened articles and resources for inclusion and data was extracted using a form created by the authors in Covidence.

### Results

The search yielded 2,049 results, with nine studies meeting the inclusion criteria. Studies varied in design, including feasibility, pilot, observational, and pre/post-intervention evaluations. The majority were conducted in European countries, with sample sizes ranging from 12 to 96 patients receiving home IV diuretics for HF. Key implementation challenges include appropriate patient selection, IV cannulation competency of healthcare providers, and multidisciplinary and multi-agency collaboration.

**Data Availability Statement:** All relevant data are within the paper and its Supporting Information files.

**Funding:** The author(s) received no specific funding for this work.

## Conclusions

Evidence on home IV diuretic administration practices for HF management remains limited. However, this scoping review suggests that commonalities across studies could form the basis for developing standard protocols in outpatient settings. Despite the lack of formal evidence-based guidelines, the findings also suggest that tailored, community-specific approaches and safe infusion guidance documents could enhance the efficacy and scalability of home IV diuretic therapy. Future research should focus on refining these strategies and exploring diuretic escalation methods beyond traditional acute care administration to optimize patient outcomes.

## Introduction

Heart failure (HF) poses a significant global health burden, with high rates of hospital readmission and mortality affecting patients around the world [1]. Despite advances in treatment, many patients experience frequent hospitalizations and declining quality of life, highlighting the urgent need for improved HF management strategies across counties and health systems [1–3]. The course of chronic HF is troubled by recurrent exacerbation episodes and an overall decline in health status, resulting in frequent hospitalizations and poor quality of life [1, 2] Loop diuretics in the form of furosemide are the treatment of choice for symptomatic relief of HF congestion [1, 3]. However, as the disease progresses, patients become less responsive to oral furosemide and require intravenous (IV) administration to address issues of diuretic resistance and cardio-renal dysfunction [4, 5].

International guidelines recommend administering IV furosemide in a hospital setting [1, 6, 7]. However, emerging literature supports alternative outpatient care pathways for HF in efforts to minimize disruption to patients and families and decentralize treatment away from hospital settings [8–11]. For instance, outpatient treatment of patients with worsening HF can avoid delays in diuretic treatment, streamline interdisciplinary HF care, and facilitate early follow-up [12]. Outpatient IV diuretic therapy refers to administration in ambulatory clinics, such as infusion clinics or day units [13]. This may also include administering IV diuretics in the patient's home. Nevertheless, there is a paucity of evidence regarding outcomes and challenges to home infusion therapy.

The scope of home infusion therapy is growing as healthcare services expand to provide comparable care to patients outside of the hospital setting. The administration of IV anti-infectives, chemotherapy, hydration, pain management, and immunotherapy is widely used in homes for managing chronic conditions such as cancer, gastrointestinal and neuromuscular and immune diseases [14–18]. The emergence of home-administered IV diuretics for HF signifies a major shift in treatment, offering benefits like patient convenience, cost savings, and reduced acute healthcare resources [8–10, 19]. Small-scale observational studies indicate that home IV furosemide can safely and effectively promote weight loss, alleviate congestion symptoms, and decrease emergency department visits and hospital admissions for some HF patients [20–22]. However, determining suitable candidates for outpatient parenteral diuretic treatment remains challenging, as eligibility involves factors beyond disease presentation, including comorbidities, psychosocial conditions and support systems [20].

Despite the growing adoption of home-based IV diuretic therapy, no review has summarized the factors influencing the home administration of IV diuretics in HF management.

Furthermore, no practice standards or evidence-based clinical recommendations guide its use or address operationalization challenges. This underscores the importance of understanding this practice's evolving implications, challenges, and opportunities to inform evidence-based decision-making, optimize patient care, and shape future research and policy initiatives. This scoping review aimed to examine and summarize the available literature on home IV diuretic administration for patients with HF. The specific research questions of this review are:

1. What are the key characteristics of the home IV diuretic HF intervention?

2. What are the facilitators and barriers to implementing home administration of IV furosemide in HF management?

## Methods

### Design and rationale

The scoping review methodological framework by Arksey and O'Malley [23] was used to guide this work with enhancements from the works of Levac et al. [24] and Peters et al. [25]. This design was chosen as it allows for an exploratory, inclusive approach to systematically map the breadth of current evidence across various study types and sources [23, 26]. Reporting adheres to the Preferred Reporting Items for Systematic Reviews and Meta-Analyses extension for Scoping Reviews (PRISMA-ScR), and the corresponding PRISMA-ScR Checklist is available in the S1 File [27, 28].

**Search strategy.** A comprehensive search strategy aimed to identify full-text publications on adult patients (age $\geq$ 18 years) with HF receiving IV diuretics in their homes. Six health databases, including the Cumulative Index to Nursing and Allied Health Literature (CINAHL), Cochrane Database of Systematic Reviews, EMBASE, MEDLINE, PsychINFO and Scopus for full articles written in the English language. A search example of Medline is included in the S2 File. All database search results were exported using the reference manager Zotero™ and imported to Covidence™, a web-based collaboration software platform. A manual search for relevant references from abstract-only sources was conducted for completeness. Finally, a search of grey literature was conducted using the same keywords within Google Scholar and combined using search modifiers. No date limit was used to capture the historical use of IV diuretics in home HF management. Database searches were conducted on April 13, 2024.

**Eligibility criteria.** The inclusion criteria was established according to the participants, concept and context of the scoping review research question and is as follows [25]:

- Full-text articles available in the English language;

- Adults (age $\geq$ 18 years);

- Patients with a diagnosis of HF (as defined within each article), regardless of etiology or ejection fraction;

- Patients receiving IV diuretic administration in the home setting (e.g., where the patient lives, such as their apartment, house or a relative's home). Home is defined as where the patient lives, such as their apartment, house, or a relative's home (Montana Code Annotated, 2021).

Grey literature that provided information on the process or effect of the intervention was included.

Articles were excluded if they were:

- Abstracts, reviews, protocols, book chapters, editorial letters, dissertations, or websites;

- Administration of IV diuretics to patients living in long-term care homes or equivalent (e.g. skilled nursing facilities) or in outpatient settings, such as infusion clinics or day units;

- Patients receiving mechanical circulatory support were excluded since this select group requires an alternative care model.

**Study selection.**    Once duplicates were automatically removed by Covidence, study screening was conducted by two independent reviewers (MK and KH). Screening criteria were piloted on 3 articles before being applied to all studies. Eligibility criteria ranking resulted in first exclusion if no full text was available (e.g. abstract only), followed by intervention type (e.g. IV diuretic delivery not in the home setting), and subsequent screening for population (e.g. adults with HF). The selection consisted of an initial title and abstract review, followed by a full-text review for inclusion and exclusion criteria. Conflicts were resolved by a third reviewer (TC) or through discussion with all reviewers until consensus was achieved.

**Data extraction.**    The research team developed a standardized data collection form through consensus guided by Sidani and Braden's intervention theory [29] to capture key information relevant to the research questions. This theory was used to detail the essential components of the IV diuretic intervention, resources needed, and the client, contextual factors and resources that influence home IV diuretic administration. Consistent with the iterative nature of scoping reviews, the charting tables were continually updated as the review team became familiar with source results [23]. The following data were collected from each study: author information, publication year, country, study design, study aims, and details about participants and IV diuretic administration events. The extraction tables were piloted on three articles by two independent reviewers (MK and KH) and reviewed by the team before finalizing. Two researchers (MK and SP) independently coded data and completed the Covidence extraction.

**Collating, summarizing and reporting the results.**    The general characteristics of the included studies were analyzed using descriptive statistics to identify trends or patterns. A descriptive numerical summary is provided on the number of studies included, types of study design, years of publication, characteristics of the study populations, and countries where studies were conducted.

To address the first research question, deductive content analysis used components of the intervention theory, as defined by Sidani and Braden [29], as initial codes to categorize and organize extracted information on home IV diuretics. This textual analysis facilitated the examination and structuring of the narrative data describing home IV diuretics' characteristics, such as dose, equipment and administration processes. Results from the charting data were collated and further described in the results section.

To address the second research question, articles were read to identify supportive strategies and barriers to home IV diuretic use. Findings from the process were organized according to human, design, and system themes, influenced by the National Coordinating Council for Medication Error Reporting and Prevention (NCC MERP) taxonomy [30, 31]. The NCC MERP taxonomy was utilized for its comprehensive and structured approach to categorizing factors influencing safe medication administration [31]. A formal methodological quality assessment for the included studies was not undertaken since scoping reviews are intended to provide a broad synopsis of the existing evidence, regardless of the methodological quality [25].

# Results

## Literature search

A comprehensive search of electronic databases yielded 2,049 results, and an additional two resources were identified through grey literature and reference list searches. Following the removal of duplicates, title and abstract screening, 99 full-text studies were assessed for eligibility. Nine articles published from 2005 to 2023 were included in the final sample. The PRISMA flow diagram (Fig 1) presents a summary of the literature search.

## Study characteristics

The included studies' general characteristics are outlined in Table 1. The included articles ranged from 2005 to 2023, with more than half of the studies (5; 56%) published in 2020 and later.

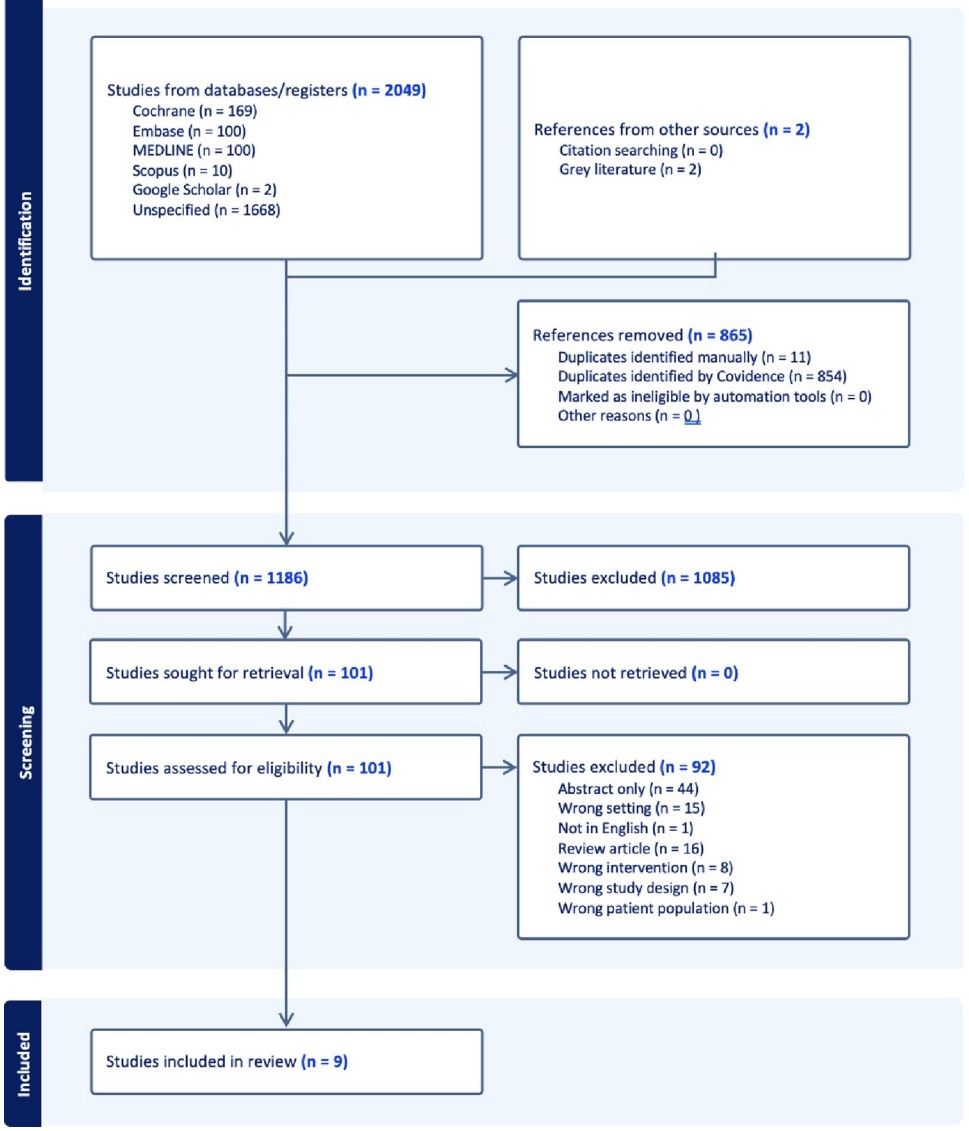

**Fig 1. PRISMA flow diagram.**

**Table 1. Characteristics of included studies.**

| First author and publication year | Country of study | Study Aim | Study Design | Number of patients who received home IV diuretics / total participants | Number of home IV diuretic administration episodes | Outcomes measured |
|---|---|---|---|---|---|---|
| Ahmed et al., 2021 | United Kingdom | To establish the feasibility, safety, and efficacy of outpatient intravenous (IV) diuretic treatment for the management of HF. | Retrospective single-site observational analysis | 33 / 79 (31.6%) | 36 | 1. Duration of IV diuretic treatment<br>2. Uncomplicated treatment outcome (re-compensation of decompensated patient, where treatment delivered entirely in outpatient setting).<br>3. Complicated treatment outcome: treatment complicated by i) AKI stage 1, defined as Cr > 1.5-fold two-fold from baseline, or ii) hospitalization was necessary prior to end of treatment episode for decompensation requiring higher dose of diuretics than could be administered in ambulatory setting or due to alternative medical problem<br>4. HF hospitalization and death after index treatment episode<br>5. Use of home care service. |
| Austin et al., 2013 | United Kingdom | Investigate the use of existing evaluation tools to monitor treatment in terms of patient nutrition and symptom changes, carer wellbeing, and satisfaction with the community diuretics service. | Pre-/post-intervention evaluation | 14 / 25 (56%) | 187 (including IV and SC) | 1. Effect of diuretic treatment on calf circumference<br>2. Effect of diuretic treatment on patient nutrition by MNA<br>3. Effect of diuretic treatment on symptom assessment by ESAS<br>4. Effect of diuretic treatment on carer wellbeing<br>5. Effect of diuretic treatment on BNP test results<br>6. Qualitative patient satisfaction |
| Brightpurpose, 2014 | United Kingdom | Evaluate if home IV diuretic administration is clinically effective, safe, improve the patient and carer experience, and is cost effective. Additionally examined sustainability of providing IV diuretics in community, and impact on improving knowledge and skill of patients, carers and healthcare providers. | Pilot study | 83 / 96 (86%) | 105 | 1. Clinical effectiveness: reduction in weight, edema, NYHA status, patient reported symptoms, renal function.<br>2. Home IV diuretic safety<br>3. Patient and carer experience and satisfaction |
| Feldman et al., 2022 | United States of America | To investigate the feasibility of integrating community paramedics into the outpatient management of patients with HF with scheduled and, if needed, urgent "house calls". | Feasibility study | 3 / 40 (7.5%) | 4 | 1. Incidence of 30-day HF readmissions<br>2. 30-day all-cause readmissions, emergency room evaluations<br>3. Unplanned office encounters and any adverse events were prospectively documented.<br>4. Patient, physician, and nurse practitioner perceptions of the MIHP process of care. |

*(Continued)*

**Table 1.** (Continued)

| First author and publication year | Country of study | Study Aim | Study Design | Number of patients who received home IV diuretics / total participants | Number of home IV diuretic administration episodes | Outcomes measured |
|---|---|---|---|---|---|---|
| Godino et al., 2022 | Italy | Evaluate the efficacy and safety of long-term home administration of diuretics via PICC in end-stage HF in patients refractory to standard therapies, such as a palliative care. | Retrospective single-centre study | 41 / 41 (100%) | Not reported | 1. Number of HF hospitalizations in the short (1–3 months), medium (6 months) and long term (1 year), before and after PICC implantation. 2. Changes in laboratory and echocardiographic parameters, weight, and NYHA class after PICC implantation. 3. Assessment of PICC's complications such as infections, thrombosis, or bleeding at the PICC insertion site 4. Economic impact of this clinical management of end-stage HF patients. 5. Number of pre- and post-PICC acute decompensated HF free days |
| Holdgaard et al., 2005 | Denmark | To gain practical experience with the treatment and logistical problems that arise in connection with (HF) home visits to better establish a permanent arrangement. | Pilot study | 3 / 12 (25%) | Not reported | 1. Number of patients transported to hospital 2. Time (in minutes) of preparation, transportation, and appointment with patient 3. Cost per visit 4. Patient and caregiver satisfaction |
| Severson et al., 2023 | United States of America | Implementation of a CP MIHP for decompensated HF requiring intermediate acuity care to increase home time | Pilot study | 24/31 (77%) | Not reported | 1. Number of inpatients readmitted within 30 days 2. Number of outpatients admitted 3. Mean change in weight in pounds (lbs) 4. Mean change in creatinine 5. Mean change in potassium 6. Mean change in sodium 7. Mean 30-day home time in days 8. All-cause mortality. |
| Van Ramshorst et al., 2022 | Netherlands | Test feasibility of providing hospital care at home, combining financial budgets, increasing workforces by combining teams, and improving perspectives and increasing patient and staff satisfaction. | Feasibility study | (n/a) / 16 | Not reported | 1. Patient satisfaction 2. Team satisfaction 3. Surrogate cost measure: number of hospital admission days saved 4. Surrogate cost measure: car rental days saved 5. Resolution of dyspnea and/or edema 6. Weight loss 7. Safety outcome: hypokalemia 8. Safety outcome: worsening renal function |

**Table 1.** (Continued)

| First author and publication year | Country of study | Study Aim | Study Design | Number of patients who received home IV diuretics / total participants | Number of home IV diuretic administration episodes | Outcomes measured |
|---|---|---|---|---|---|---|
| Veilleux et al., 2014 | United States of America | To demonstrate that augmented diuretic therapy, both oral and intravenous, an evidence-based treatment for care of patients with HF experiencing fluid retention, can be delivered safely in the home setting using the HDP and can improve outcomes for recently hospitalized patients with HF. | Feasibility study | 10 / 60 (16.7%) | 13 times within 10 activations, 3 patients received IV diuretics on 2 consecutive days | 1. Home diuretic protocol (HDP) activation and adherence 2. Patient response measured by physical findings of change in baseline weight and vital signs 3. Patient response measured by symptomology (e.g. dyspnea, orthopnea, rales, peripheral edema, early satiety, abdominal bloating, chest pain) 4. Changes in laboratory values, including serum creatinine and electrolytes 5. Patient disposition after each episode of care 6. Patient and clinician satisfaction 7. Hospital readmission rate 8. Safety and feasibility of IV diuretic administration |

Acute kidney injury (AKI); Brain natriuretic peptide (BNP); Chronic kidney disease (CKD); Creatinine (Cr); Community Paramedic (CP); Edmonton Symptom Assessment System (ESAS); Emergency department (ED); Guideline directed medical therapy (GDMT); Heart failure (HF); Intravenous (IV); Internal cardiac defibrillator (ICD); Mini Nutritional Assessment (MNA); Mobile integrated health paramedic (MIHP); Myocardial infarction (MI); New York Heart Association status (NYHA); Peripherally inserted central venous catheter (PICC); Safety and effectiveness of Acute heart Failure care as outpatient Randomized Controlled Trial (SAFE-RCT); Subcutaneous (SC).

Most studies were conducted in European countries (6; 67%) [32–37], and three (33%) were conducted in North America [38–40]. In terms of the study design, three (33%) were feasibility studies [38–40], three (33%) pilot studies [34, 36, 39], two retrospective single-site observational studies [35, 41], and one (12.5%) pre/post-intervention evaluation [33].

The administration of IV diuretics in the home was the primary study aim in four of the nine studies (44%) [32, 34, 35, 40], whereas the others evaluated home IV diuretics within a program focused on symptom management [35] or home HF visiting programs by nurses or community paramedics [36–39]. Two (25%) studies reported mixed intervention groups (N = 121), including both IV and subcutaneous (SC) diuretic administration [33, 34].

A total of 211 participants were included across the 9 studies, with a range of sample sizes from 12 to 96 (mean 45) [32–40]. The age of patients included in the studies ranged from 37 to 90 years, with a mean age of 75 [34–42]. Patient sex was reported in seven (78%) of the nine studies [32–35, 37–39]. The percentage of female patients ranged from 14% to 50%, with a total of 108 female patients reported across all studies [32–35, 37–39]. One (12.5%) study reported ethnicity, where 24.1% were non-white ethnic minorities [32].

All studies included adult patients with a diagnosis of HF who failed to respond to increasing doses of oral diuretic treatment residing within a pre-specified catchment area [32–40]. Five (56%) studies recruited patients during their hospital admission [32, 37–40]. New York Heart Association (NYHA) and left ventricular ejection fraction (LVEF) criteria varied across studies and are further detailed under 'Client Factors'. Most articles (n = 6; 67%) excluded

patients who demonstrated hemodynamic instability due to hypotension or metabolic and electrolyte abnormalities [32–34, 37–39].

Due to heterogeneity in the reporting of NYHA and LVEF, the authors were unable to present an accurate categorical reflection of the patient population. Overall, five (56%) studies reported NYHA status II-IV indicating treatment of symptomatic HF [33–36, 39]. There was a lack of standardized LVEF reporting structure according to international guideline-recognized phenotypes. In the six (67%) studies that reported LVEF, home IV furosemide was used across all LVEF subtypes and was not a consideration for treatment decisions [32, 34, 35, 37–39]. Patients included across all studies presented with a variety of co-morbid conditions (e.g., ischemic heart disease, hypertension, atrial fibrillation, renal dysfunction, lung disease, diabetes mellitus, anemia, frailty). Reporting of co-morbidities across studies was inconsistent and, therefore, difficult to summarize within the final sample. Additional information regarding co-morbidities of each study are available in Table 2.

## The intervention

Sidani and Braden's [29] intervention theory helps describe the intervention at conceptual and operational levels. Conceptually, it outlines the intervention's overall goal and active components. Operationally, the theory guides how elements are translated into specific components, detailing the content, activities, mode of delivery, timing, sequence, and dose. These components as described by authors are summarized in Table 3.

**Goal of home IV diuretics.** The general goal of home IV diuretic therapy is to achieve symptom relief from fluid congestion and avoid acute care utilization. The articles present that this approach can be further differentiated into remedial approaches that target active management and restoration of euvolemia, versus palliative treatment measures, focusing on comfort and quality of life. Six (67%) studies administered home IV diuretic therapy for acute management of decompensated HF episodes [32, 36–40], while two (22%) studies administered it solely for palliative care purposes [33, 35], and one study combined both indications [34].

**Medication mode of delivery.** Furosemide was the diuretic of choice in all studies [32–40]. Administration of the IV furosemide was specified by peripheral IV (PIV) catheter in five studies (55%) [32–34, 38, 39] and by peripherally inserted central venous catheter (PICC) in two studies (22%) [34, 35]. The medication was delivered via continuous infusion in four studies (44%) [32, 35, 38, 40] compared to bolus dosing in two studies (12.5%) [33, 34].

**Dose, timing and sequence.** Five articles (56%) reported on daily IV furosemide doses [32–35, 39]. The majority (n = 4, 44%) used doses ranging from 40 mg to 250 mg [32–34, 37, 39], though Godino et al. [35] reported a maximum daily dose of 480 mg (mean 302.5 mg) in the setting of long-term (average 271 days) IV infusion by PICC for end-stage HF management. Otherwise, the length of home IV diuretic treatment ranged from 1 to 17 days, with most studies (56%) reporting a 7-day average length of treatment [32–34, 37, 39]. Intravenous diuretic administration frequency was daily in four studies (45%) [32, 33, 35, 40], up to twice a day in three studies (33%) [34, 37, 39], and up to three times a day in one study (11%) [36]. One study described weekday-only administration [40] and two studies supported 7-days per week service [32, 39].

**Medication administration activities.** Diuretic protocols are treatment algorithms for flexible diuretic dose titration based on a patient's response to diuretics [42, 43]. Automated diuretic dose augmentation can occur when signs and symptoms of worsening congestion are present, such as body weight gain, lack of HF symptom improvement, or worsening peripheral edema [33, 40]. They promote timely dose adjustment without the need for physician reassessment. Examples of diuretic protocols were used in four studies (44%) [33, 34, 38, 40], and one

**Table 2. Characteristics of patients.**

| Study | Population Details | Inclusion Criteria | Exclusion Criteria | Clinical Characteristics | Co-morbidities (%, where available) |
|---|---|---|---|---|---|
| **Ahmed et al., 2021 (UK)** | Adult patients with decompensated chronic HF, either community-based or inpatient meeting specified criteria. | 1. HF nonresponsive to diuretic escalation. 2. Inpatients on IV diuretics willing to complete outpatient treatment. | 1. Hemodynamic instability. 2. Arrythmias causing decompensation. 3. Co-existing acute illness requiring admission. 4. Severe valvular disease. 5. Unfeasible community support. | Mean age: 77 years (49–93). Female: 34 (43%). LVEF < 40% (14, 43.8%). NYHA not specified. Home status not described. | CKD (10, 31.3%), IHD, AF COPD, DM, HTN |
| **Austin et al., 2013 (UK)** | Community dwelling patients under cardiology care for advanced HF with fluid overload. | 1. NYHA IIIb/IV with HF diagnosis. 2. Documented preference for home/ nursing care. 3. Appropriate IV/SC venous access. | 1. Hemodynamic instability. 2. Non-HF causes of fluid overload. 3. Severe CKD (creatinine > 250μmol/L). 4. Severe electrolyte disturbances: sodium < 125mmol/L, potassium < 3.5mmol/L or >5.5mmol/L. 5. Inadequate social support or severe dementia. | Mean age: 70 years (44–88). Female: 2 (14%). NYHA III: 9 (64%); IV 4 (31%) Caregiver at home: 100% | Frailty, low body mass index, neuro-psychological problems, malnutrition |
| **Brightpurpose, 2014 (UK)** | Community-based HF patients registered with a nurse service. | 1. Confirmed HF diagnosis. 2. Living in the community. 3. Appropriate nurse service. | 1. Hemodynamic instability. 2. Poor venous access. 3. Co-existing acute illness: infection, chest pain 4. Severe valvular disease. 5. Unfeasible community support. 6. Physician preference. | Mean age: 75 years Female: 34 (24%) NYHA II 4 (3%); II-III 5 (4%); III 70 (56%); III-IV 20 (16%); IV 26 (21%) LVEF not specified. Living alone 27%, 73% with caregivers. | IHD, AF, HTN, CKD, DM, anemia, pulmonary disease, valvular dysfunction, bradycardia |
| **Feldman et al., 2022 (USA)** | Patients admitted with HF across NYHA stages and LVEF ranges meeting advanced care criteria. | 1. NYHA II-IV. 2. Signs of decompensated HF with either reduced or preserved LVEF. Geographic residence within care radius. | 1. Severe valvular or pulmonary disease. 2. Dialysis dependence. 3. Severe cognitive impairment or low survival probability (6-12months). | Mean age: 68 years (37–90). Female 10 (25%) LVEF ≤ 40% (70%); > 40% (30%). NYHA not fully detailed. Home status not described. | DM, AF, dyslipidemia, CKD, liver failure, vascular disease. |
| **Godino et al., 2022 (Italy)** | End-stage HF patients with compliance to diuretic therapy and family support, requiring long-term IV care. | 1. Recurrent decompensation despite optimized therapy. 2. LVEF < 35% or HF symptoms with no transplant eligibility. 3. Proper family and compliance support. | 1. PICC inserted but never used. 2. Limited PICC use (< 1 week). 3. PICC used only during hospitalization. | Mean age: 81 years. Female 13 (32.5%). NYHA III-IV. LVEF 35 ± 16%. Home status not described. | CKD: 80%, CAD, DM, COPD, AF, previous CABG, PCI or MI. |
| **Holdgaard et al., 2005 (Denmark)** | HF patients receiving joint care from homecare and hospital services. | 1. Patients requiring HF monitoring and diuretic titration. | Not explicitly defined; exclusion primarily based on clinical judgement. | Mean age not specified. NYHA stages and LVEF note detailed. Home status not described. | Not detailed. |
| **Severson et al., 2023 (USA)** | HF patients requiring IV diuresis with access to paramedic services. | 1. Stable HF requiring IV diuresis continuation. 2. Adequate home safety and caregiver support. | 1. Severe electrolyte disturbances: sodium < 125mEq/L, potassium < 3.0 mmol/ or > 6.0mmol/L, magnesium < 1.8mg/dL. 2. Active substance abuse or behavioral health diagnosis. | Mean age: 73 years (48–87) Female: 7 (23%) NYHA III (87%), IV (10%) Mean LVEF 53% (18–66%); HFpEF 27 (87%), HFrEF 4 (13%). Home status not described. | DM, HTN, CAD, AF, COPD. |

*(Continued)*

**Table 2.** (Continued)

| Study | Population Details | Inclusion Criteria | Exclusion Criteria | Clinical Characteristics | Co-morbidities (%, where available) |
|-------|-------------------|-------------------|-------------------|-------------------------|-------------------------------------|
| **Van Ramshorst et al., 2022 (Netherlands)** | HF patients supported through an integrated teaching hospital and homecare program. | 1. HF with either reduced or preserved EF. 2. Patients requiring monitoring or treatment escalation. | 1. Early change to oral therapy (n = 34). 2. Need for inotropy or invasive cardiac studies (n = 18). 3. No patients included on Fridays or weekends. | Mean age 78 years (70–81). Female 5 (50%). NYHA not specified. LVEF according to HF classification: HFpEF 8 (50%), HFmrEF 4 (25%), HFrEF 4 (25%). Home status not described. | Frailty, HTN, DM, previous MI/CABG/PCI, AF, CKD. |
| **Veilleux et al., 2014 (USA)** | HF patients enrolled in a homecare nursing program utilizing telehealth at discharge. | 1. Patients hospitalized with advanced HF (NYHA III/IV). 2. Eligible for homecare and telehealth monitoring services. | Not reported. | Not described. | DM, pneumonia, CKD. |

Atrial fibrillation (AF); Chronic kidney disease (CKD); Chronic obstructive airways disease (COPD); Coronary artery bypass grafting (CABG); Coronary artery disease (CAD); Diabetes mellitus (DM);); Ischemic heart disease (IHD); Heart failure with preserved ejection fraction (HFpEF); Heart failure with moderately reduced ejection fraction (HFmrEF); Heart failure with reduced ejection fraction (HFrEF); Hypertension (HTN); Left ventricular ejection fraction (LVEF)); Myocardial infarction (MI); New York Heart Association status (NYHA); Percutaneous coronary intervention (PCI); Peripherally inserted central venous catheter (PICC); Systolic blood pressure (SBP); United Kingdom (UK); United States of America (USA)

study (12.5%) outlined a decision pathway for treatment at home versus day unit according to renal function [32].

Treatment monitoring parameters were reported in seven studies (78%) [32, 34, 36–38, 40]. Standard parameters included bloodwork for electrolytes and renal function, weight, vital signs, and symptomology (NYHA status) (n = 6, 67%) [32–34, 36, 37, 40]. Others included complete blood count (n = 1, 11%) [33], electrocardiograms (n = 1, 11%) [36], phlebitis scores (n = 1, 11%) [33], and subjective scores for care satisfaction and caregiver burden (n = 2, 22%) [33, 34]. Three studies (33%) described virtual services, including remote home monitoring systems for weight and vital signs and telehealth services to connect in-home providers with HF specialists [32, 38, 40]. One study (11%) described follow-up care, requiring patients to attend an in-person outpatient HF clinic 30 days following therapy initiation [37].

Other administration activities included medication verification and documentation. One study (11%) described medication verification by two nurses before delivery [34], while another (11%) performed verification by mailing a photo to the central office [37]. Documentation of therapy by the health care provider was described in three studies (33%) [32, 36, 38]. Two studies (22%) utilized charts left in the patient's home [36, 38], and one (11%) used a standardized electronic template uploaded to the electronic medical record [32].

## Patient factors

Patient factors are the personal and health or clinical characteristics that may influence the intervention [29]. Patients were excluded from studies based on clinical, psychosocial, or resource-related factors which may have influenced the effectiveness of the IV diuretic program. Clinical factors that resulted in the exclusion of patients from home IV diuretic programs include hemodynamic instability (e.g. systolic blood pressure less than 90 mmHg, tachycardia due to uncontrolled arrhythmia, or hypoxia), metabolic abnormalities (e.g. hyponatremia, hypo- or hyperkalemia, elevated creatinine indicative of acute kidney injury or on

**Table 3. Home intravenous diuretics for heart failure process described by authors.**

| First author, Year | Ahmed et al., 2021 | Austin et al., 2013 | Bright purpose, 2014 | Feldman et al., 2021 | Godino et al., 2022 | Holdgaard et al., 2005 | Severson et al., 2023 | Van Ramshorst et al., 2022 | Veilleux et al., 2014 |
|---|---|---|---|---|---|---|---|---|---|
| Prescriber type | Physician; acute HF specialist with on-call physician | Physician | Physician | HF specialist in conjunction with on-call command physician | Not specified | Physician | Physician or Advanced practice provider (APP) / Nurse | Physician | Physician, cardiologist and primary care |
| Type of healthcare provider administering IV diuretic | IV adult community therapy nurse | Community HF specialist nurse (CHFSN) | Nurse | Mobile Integrated Health Paramedic (MIHP) | Non healthcare provider caregiver or family | Home HF nurse | Mobile Integrated Health Paramedic (MIHP) | Nurse | Home health nurse |
| Education and training | Usual skillset, no additional training | Not described | Training and practice for nurses on PIV cannula insertion, IV diuretic administration. Project implementation training to stakeholders and administrators. | 2-day HF curriculum training, including mock drills in classroom and private residence, in addition to time spent with a specialized HF nurse practitioner in HF clinic | Patients and family members educated on how PICC works, rational on decision to implant, and use and cleaning for correct functioning | 3-day course on diagnostics and treatment of patients with HF organized by the Danish Nursing Council and Danish Cardiological Society | Not described | Homecare nurses trained to give IV diuretics during few days on clinical ward and hospital nurse practitioners trained in home visits by homecare nurses | Education and training to support project implementation to inpatient nurses, discharge planners, cardiologists, hospitalists, home health nurses, and primary care physicians. Specific training of homecare nurses on HF clinical updates and implementation of HDP. |
| IV diuretic administration method | Infusion by PIV catheter | Very slow bolus by PIV catheter | Bolus 123 (98%) interventions, Infusion 3 (2%); 121 (96%) by PIV, 3 (2%) by PICC, 2 (1%) by midline | Infusion by PIV catheter | Infusion by PICC | Not specified | Via PIV | Not specified | Infusion by non-specified access or equipment |
| IV diuretic interval or frequency | Daily, 7-days per week | Daily | Once daily 83 (66%); Twice daily 41 (33%); Started as twice and moved to once daily 2 (2%). Sites varied, some supported 5-day service where patients returned to oral diuretics over weekend and others supported 7-days a week | Not specified | Daily | Not specified, though nurse could visit up to 3-times per day | Daily to twice a day | Twice a day | Daily, Monday to Friday |

(*Continued*)

**Table 3.** (Continued)

| First author, Year | Ahmed et al., 2021 | Austin et al., 2013 | Brightpurpose, 2014 | Feldman et al., 2021 | Godino et al., 2022 | Holdgaard et al., 2005 | Severson et al., 2023 | Van Ramshorst et al., 2022 | Veilleux et al., 2014 |
|---|---|---|---|---|---|---|---|---|---|
| Average length of IV diuretic treatment, range | 7 days (1–14 days) | ≤7 days (1–14 days) | 7 days, 1–32 days | Not specified | 271 days | Not specified | 4.94 days (1–13 days) | 11 days +/- 6, total 201 days | Not specified |
| IV diuretic administration time | Up to 60 minutes | Not specified | Not specified | Not specified | Not specified | Not specified | Not specified | Not specified | Not specified |
| IV diuretic total daily dose | Furosemide, maximum 240 mg | Furosemide, range 40 mg 160 mg | Furosemide, range 40–250 mg | Not specified | Furosemide, avg. 302.5mg +/- 176mg | Not specified | Furosemide, avg. 143.2mg (20-240mg) | Not specified | Not specified |
| Treatment monitoring | Serum electrolytes, creatinine and hepatic profile, full blood count; vital signs and weight; and symptomology. | Phlebitis score, serum electrolytes, Edmonton Symptom Assessment System, Carer's stress scale, Mini nutritional assessment, service satisfaction. | Serum electrolytes and creatinine, vital signs, weight, and waist circumferences, symptomology, fluid balance. | Post administration monitoring for 60 minutes for acute effects. | Not specified | Serum samples not specified; vital signs, weight; symptomology and functional level. | Not specified | Serum samples not specified, electrocardiogram, vital signs, weight, symptomology. | Serum electrolytes, creatinine, magnesium; physical assessment including vital signs, weight; symptomology. |
| Follow-up | Not specified | Not specified | Not specified | Not specified | 1-year clinical follow-up available for 33 patients | First visit within 1-week of discharge | Not specified | In-person outpatient HF clinic visit at 30 days. | Not specified |
| Documentation | Chart left in patient home | Not specified | Not specified | Standardized electronic template in electronic medical record | Not specified | Home care communication book in patient's home | Not specified | Not specified | Not specified |
| Diuretic pathway or protocol described | Yes | Yes | Yes | Yes | Not specified | Not specified | Not specified | Not specified | Yes |
| Use of remote home monitoring system or telehealth services | Yes | Not specified | Not specified | Yes– videoconferencing | Not specified | Not specified | Not specified | Not specified | Yes |

Emergency department (ED); Heart failure (HF); Home diuretic protocol (HDP); Intravenous (IV); Mobile integrated health paramedic (MIHP); New York Heart Association status (NYHA); Peripherally inserted central venous catheter (PICC); Peripherally inserted venous (PIV) catheter

dialysis) or who were experiencing worsening HF because of another acute medical illness that required treatment, such as worsening chronic obstructive pulmonary disease, renal failure, or anemia [32, 33, 37–39]. Most articles (n = 6, 67%) excluded patients with hemodynamic instability due to hypotension or metabolic and electrolyte abnormalities [32–34, 37–39]. Patients were excluded in five studies (56%) if they had inadequate social support or were perceived to have unsafe home environments [32–34, 38, 39]. One study (11%) cited limited staff coverage and excluded patients if they required weekend care [37].

**Contextual factors.** Contextual factors are defined as the physical, psychosocial, and political features of the setting in which health professionals deliver an intervention and the environment or life circumstances in which clients apply treatment recommendations, impacting the intervention's optimal implementation and ultimate outcomes [31]. Two (25%) studies reported on patients' living status, with 73% (N = 70) to 100% (N = 25) of patients living with a caregiver or family [33, 34]. Contextual factors such as home safety with access to electricity, water and heat, and securement of pets were noted in the inclusion criteria of two (22%) studies [33, 39]. Details regarding patients' living environment, such as whether they lived in a house, apartment, or urban versus rural locations, were unavailable. Factors such as education, employment, income, social support and security, partner status, and diagnosis of depression or anxiety have been shown to impact morbidity, mortality, hospital readmission rates, pharmacotherapy adherence, and the cost of care of patients living with HF [44, 45]. These are known to be interrelated psychosocial factors that significantly impact HF-related outcomes but were not outlined in the included review articles.

**Resources.** Resources are the material and human assets necessary for delivering home IV diuretics [29]. Material resources include cannulas and dressings, IV bags and tubing, PICC accessories (e.g., dressings), blood draw equipment, and monitoring equipment like vital sign machines and scales [32–40].

Human resources encompass all personnel involved, such as clinician prescribers and IV furosemide administrators whose qualifications and attributes significantly impact the intervention's implementation and effectiveness [32, 34, 36–40]. Eight studies (89%) specified physicians as the prescriber for home IV diuretic therapy [32–34, 36–40], with four (44%) describing shared prescribing between HF specialists, emergency department and primary care [32, 38–40], and two (22%) involving a non-physician prescriber, such as an advanced practice nurse (APN) [39, 41].

Home care nurses were the predominant providers for home IV diuretic administration [32–34, 36, 37, 40]. Two studies (22%) described utilizing paramedics [38, 39], and one study (11%) [35] involved a non-healthcare provider, caregiver, or family member administering IV diuretics. Notably absent in the literature was the role of the pharmacist.

**Facilitators and barriers.** Facilitators and barriers related to home IV diuretic administration were organized at the human, design, and system levels [31], all of which could influence the effectiveness of the intervention. These are presented in Table 4. Human-level factors were identified based on how authors described the roles, responsibilities, and interactions of healthcare professionals, patients, and caregivers in managing home IV diuretic therapy. Common human-level facilitators include clinical expertise in HF, multidisciplinary engagement, and professional education and training [32, 34, 36–38, 40]. Education for home care providers was described in studies to focus on HF care (n = 7, 78%) [32, 34–38, 40], and IV cannulation and medication administration (n = 5, 56%) [34–38, 40]. Human-level barriers included provider safety concerns about medication side effects in the home setting, inappropriate patient selection, staffing limitations to meet the needs of service delivery, and poor communication between the prescriber and the medication administer which led to care coordination issues [32–34, 36, 38–40].

**Table 4. Home IV diuretic administration facilitators and barriers.**

| Author, Year | Facilitators | | | Barriers | | |
|---|---|---|---|---|---|---|
| | Human | Design | System | Human | Design | System |
| **Ahmed et al., 2021** | Greater patient autonomy through home support networks. | HF hospital-based team supervision. | Multi-agency approach to care delivery. | Provider saftery concerns regarding hypotension or renal dysfunction. | Higher-risk patients preferred day-unit review by HF specialist. | Not described. |
| **Austin et al., 2013** | Reduced stress/ negative feelings of caregivers. Appreciateion by patients for home IV therapy. | HF nurses access and palliative care team at each site. | Not described. | Difficulty maintaining IV cannulation skills. Staff capacity issues with IV access. | Lack of staffing resources to meet demand. | Not described. |
| **Brightpurpose, 2014** | Training nurses for HF and IV skills. Recruiting experienced pilot leads. Patience during 6-month set up period. | Midline insertion for secure IV access. Simple prescribing processes to reduce clinician workload. | Collaboration with local services for weekend on-call rotations. Sharing lessons via learning events. | Staff availability issues. Resistance to multiple nurses administering medication. | Limited infrastructure for 7-day service. Equipment and supply issues. | Fragmented services and lack of cross-discipline buy-in. |
| **Feldman et al., 2022** | HF-specific training for MIHP. | Rapid triage and monitoring post-IV administration. Follow-ups for consecutive interventions. | Partnerships between on-call physician, HF specialist, and MIHP. | Suboptimal clinical skills and poor verbal/ telecommunication pathways. | Cost of additional MIHP training. | Poor integration with hospital networks and EMRs. |
| **Godino et al., 2022** | Patient/caregiver education on PICC care before discharge. | PICC lines simplified home care management. Continuous phone/ ambulatory support. | Not described. | Not described. | Not described. | Not described. |
| **Holdgaard et al., 2005** | Passive/active HF education for support workers. | Collaborative visits between HF and homecare nurses | Not described. | Unclear nurse roles. Reluctance for hospitalization among patients. | Physician diengagement from projects requiring outpatient follow-ups. | Not described. |
| **Severson et al., 2023** | HF-specific training adapted from credible sources. | Simple EMR reduced data fragmentation. Access to paramedics 7-days per week for care. | Multi-disciplinary taskforce reviewed safety and adjusted workflows bi-weekly. | Limited paramedic training (only two staff). | Capacity limited by staff, lab processing times, and travel constraints. | Lack of funding. Voluntary participation from staff. |
| **Van Ramshorst et al., 2022** | Nurse practitioners trained by cardiologists; high staff satisfaction. | Integrated budgets of hospital and primary homecare groups. | Information technology department involved in EMR communication development. | Not described. | Not described. | Not described. |
| **Veilleux et al., 2014** | Committed engagement by home health leadership. Competency of trained telehealth nurses. | Use of telemonitoring for early detection of worsening conditions. Algorithm-driven diuretic administration protocol. | Multi-disciolinary taskforce engagement supported care. | Physician reluctance for new EMR. Fewer-than -expected patients recruits. | Costs associated with telehealth nursing and physician time. | Interrupted supply chains for medication and equipment. |

Electronic medical record (EMR); Heart failure (HF); Intravenous (IV); Mobile integrated health paramedic (MIHP); Peripherally inserted central venous catheter (PICC)

Design-level factors focused on structural elements that influence the delivery of home IV diuretic therapy, such as organizational strategies, care coordination approaches, and logistical considerations. These included multidisciplinary approaches to care delivery that enhanced partnerships between hospital networks, primary care and homecare services, for example, through documentation, combined visits, and case reviews [32, 34, 36, 38–40]. Supportive design factors enhanced documentation, point-of-care testing, and evaluation, ensuring

seamless communication between team members. The authors identified the most significant operational challenge as skill maintenance for IV canulation [34, 35, 37–40]. Additionally, resource constraints were identified when inadequate staffing or medication supply was interrupted [33–36, 38, 40].

System-level factors include broader healthcare dynamics, policies, regulations, and resource allocation. System-level facilitators included effective multi-agency communication, strong project leadership, and sufficient lead-in time to implementation to organize stakeholder's [32, 34, 38–40]. System barriers to home IV diuretic administration included unclear cost reimbursement strategies and a lack of sustainable program funding [34, 38, 40].

## Discussion

Home IV diuretics are used in the management of HF patients despite a lack of formal guidelines or clinical recommendations [20, 46]. This scoping review provides the most recent evidence on home IV diuretic treatment in HF patients, focusing on their key characteristics and the facilitators and barriers to its implementation. Nine studies were identified, offering insights into international home IV diuretic use from 5 countries administered to 211 participants. Commonly, home IV diuretics were used to address congestive symptoms of HF while attempting to avoid hospitalization [32–35, 40]. Studies selected adult patients with symptomatic HF of any type and co-morbid conditions whose hemodynamic and metabolic measures were stable, excluding vulnerable patients who were unlikely to cope with the added burden of parenteral care, such as increased urine output at home and infusion-related equipment [32–34, 38, 39]. Few studies detailed the socioeconomic factors essential to support patients in the home setting, and only two articles considered patient prreference for home IV therapy [32, 38].

Overall, the studies utilized IV furosemide dosing and procedures akin to standard care, often involving the administration of high doses via peripheral IVs or PICC lines in short bolus infusions [1, 6, 7, 47]. This is similar to other observational studies using IV furosemide in the outpatient clinic setting, suggesting that the standard dosing protocols are consistently applied across different clinical environments and patient populations, reeinforcing the reliability and generalizability of IV diuretic dosing [48–52]. The uniformity in dosing practices indicates a recognition and acceptance of the efficacy of these protocols despite the absence of robust randomized data.

The dosing approaches observed in these studies included a total daily dose of 40 to 240mg of furosemide, commonly administered via bolus delivery for 1 to 2 weeks [32–34, 37, 39]. Although specific dosing intervals were not always explicitly outlined, standard care practices would typically suggest twice-daily dosing for total daily amounts at or above 80mg, with higher doses up to 120mg twice a day [53]. Additionally, the study by Godino et al. reported a mean daily dose of 302.5 ± 176 mg; however, this dosing regimen was delivered through a continuous infusion pump rather than bolus injections, reflecting a variation in administration method that may impact overall efficacy and ability to support this intervention.

There are no practice standards or evidence-based clinical guidelines for using home IV diuretic therapy in HF management. This review highlights essential elements needed to develop such standards. The Gorski Model for Safe Home Infusion offers a potential framework that emphasizes five aspects of care: patient selection, comprehensive care planning (including assessment and monitoring), interprofessional communication and collaboration, patient and caregiver education, and home care organization [54–56]. Essential elements identified in this review include 1) Patient selection criteria that identify potential candidates based on both clinical and psychosocial factors; 2) The ability to obtain IV access is an essential skill of nursing and paramedicine; however, it was a rate-limiting factor in supporting home IV

diuretic therapy identified by both professions [33, 34, 38]; 3). Multidisciplinary and multi-agency collaboration facilitated by regular communication and supportive staffing structures; and, 4) Tailored approaches that are necessary to address specific local needs and challenges, such as detailed roles and responsibilities, established communication channels, and exhaustive implementation planning [57–60]. This review did not find detailed information on patient and caregiver experiences or education, though future research to address this need is critical for active participation, informed decision-making, and potentially mitigating home IV therapy risks and complications in specific patient populations [61, 62].

This review demonstrates the evolution of healthcare team models to meet evolving patient and health system needs. Novel integration of paramedics into the homecare setting poses unique opportunities and challenges. While helpful in addressing health resource issues, integrating paramedics requires careful consideration of training requirements, the scope of practice delineation from nursing, and coordination with existing healthcare teams [38, 39, 63, 64]. Surprisingly absent were primary care providers, and pharmacists, who are known to play pivotal roles in HF management [65–68]. APNs appeared in the literature less than anticipated. APNs are essential in managing heart failure, as they provide critical support in complex treatments, enhance provider-patient communication, and ensure effective symptom monitoring and response. Incorporating APNs into home IV diuretic therapy can address significant implementation barriers and support advanced homecare practices [69–74]. It is prudent for future research design and evaluation to focus on implementation frameworks that enhance the equitable, sustainable and scalable deployment of home IV diuretics across different environments.

Finally, the use of subcutaneous (SC) furosemide is growing in popularity. This evolving delivery strategy has similar bioavailability and efficacy as IV infusion, does not require cannulation, and can allow patients to self-administer and titrate [75–77]. The use of IV and SC diuretics in patients with advanced chronic HF remains an area of high priority for future research to determine its clinical and cost-effectiveness in contrast to combined oral diuretic strategies [78, 79].

## Strengths and limitations

The strengths of this review include a rigorous and reproducible search strategy guided by a librarian. The results included all available evidence in full-text on home IV diuretics for patients with HF. The scoping review methodology included grey literature and did not exclude studies based on methodological quality to provide a fulsome summary of home IV diuretic treatment for patients with HF. Although scoping reviews typically focus on information breadth rather than depth, the intervention theory used for data collection and analysis helped to structure the narrative data presented in the wide range of articles, ensuring a complete collection of relevant details and context to shape our interpretation and understanding [27].

The limitations of the review are influenced by the lack of available literature on home IV diuretics. The exclusive search for English-language articles in the search strategy may have exacerbated this. The search revealed several abstracts on home IV diuretic use for patients with HF (n = 41) that may have contributed to a broader understanding of different models of care but were excluded in the absence of full-text publication. Literature suggests that many abstracts fail to be published in total, which can lead to publication bias where positive results are more frequently published than neutral or negative results [80]. For our investigation, this may result in a misunderstanding of real-world challenges for implementing home IV diuretic programs.

The major limitation of this publication is the lack of comprehensive evaluation regarding the outcomes and effectiveness of home IV diuretic therapy for HF. Fully understanding these

outcomes is challenging due to variability in study designs, small sample sizes, and diverse patient population. The differences in home care protocols, implementation, and patient monitoring and reporting complicate the standardization of approaches for evaluation or directly compare study results. These limitations underscore the need for more rigorous studies and reporting structures to clarify the true impact of home IV diuretic therapy on long-term HF outcomes.

## Conclusion

This scoping review synthesized the literature to determine how home administration of IV diuretics impacts patients and families, healthcare providers, and the healthcare system. Home IV diuretic programs for HF aim to optimize healthcare resource utilization by shifting aspects of HF management from acute care settings to the home environment while maintaining quality of care and patient safety [32–35, 40]. Many barriers to its operationalization were identified, but none appear insurmountable if countered with strategic implementation. This area of practice continues to lack rigour compared to other cardiovascular therapeutics due to the limitations of available randomized controlled evidence [81]. Future research should validate IV home diuretic protocol and establish contemporary clinical guidance documents on developing, implementing, and evaluating this intervention.

## Supporting information

**S1 File. Preferred Reporting Items for Systematic reviews and Meta-Analyses extension for Scoping Reviews (PRISMA-ScR) checklist.**
(DOCX)

**S2 File. Sample search strategy in Ovid MEDLINE.**
(DOCX)

## Author Contributions

**Conceptualization:** Morgan B. Krauter, Katherine S. McGilton, Tracey J. F. Colella.

**Data curation:** Morgan B. Krauter, Stuti S. Patel, Karen Harkness.

**Formal analysis:** Morgan B. Krauter, Katherine S. McGilton, Karen Harkness, Tracey J. F. Colella.

**Investigation:** Morgan B. Krauter, Katherine S. McGilton, Karen Harkness, Tracey J. F. Colella.

**Methodology:** Morgan B. Krauter, Katherine S. McGilton, Karen Harkness, Tracey J. F. Colella.

**Project administration:** Morgan B. Krauter.

**Supervision:** Katherine S. McGilton, Tracey J. F. Colella.

**Validation:** Karen Harkness, Tracey J. F. Colella.

**Writing – original draft:** Morgan B. Krauter.

**Writing – review & editing:** Morgan B. Krauter, Katherine S. McGilton, Stuti S. Patel, Karen Harkness, Tracey J. F. Colella.

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
