## [Decision Letter · Decision Letter 0]

4 Nov 2024

PONE-D-24-43583Home intravenous diuretic administration for heart failure management: A scoping review.PLOS ONE

Dear Dr. Krauter,

Thank you for submitting your manuscript to PLOS ONE. After careful consideration, we feel that it has merit but does not fully meet PLOS ONE’s publication criteria as it currently stands. Therefore, we invite you to submit a revised version of the manuscript that addresses the points raised during the review process.

**ACADEMIC EDITOR: **All issues raised by expert reviewers are required.

We look forward to receiving your revised manuscript.

Kind regards,

Vincenzo Lionetti, M.D., PhD

Academic Editor

PLOS ONE

Reviewers' comments:

Reviewer's Responses to Questions

**Comments to the Author**

1. Is the manuscript technically sound, and do the data support the conclusions?

Reviewer #1: Yes

Reviewer #2: Yes

2. Has the statistical analysis been performed appropriately and rigorously? 

Reviewer #1: Yes

Reviewer #2: N/A

3. Have the authors made all data underlying the findings in their manuscript fully available?

Reviewer #1: Yes

Reviewer #2: Yes

4. Is the manuscript presented in an intelligible fashion and written in standard English?

Reviewer #1: Yes

Reviewer #2: Yes

5. Review Comments to the Author

Reviewer #1: This is a scoping review about home intravenous diuretic administration for heart failure management. The review summarized the much research about home intravenous diuretic therapy and presented very comprehensive, easy-to-understand, compact information about home intravenous diuretic therapy. I think the manuscript was finely arranged.

I have only one minor issue.

It would be more useful if there were more information on the results and effectiveness of IV diuretics treatment, if possible.

Reviewer #2: I have read with great interest the paper entitled " Home intravenous diuretic administration for heart failure management: A scooping review." The paper is generally well written and addresses a significant clinical issue. Presenting the majority of the results in tabular form is an effective approach that provides a clear and concise overview of the presented data.

I have few minor comments:

1) The introduction is deficient in terms of global data, as it presents only the data from Canada as representative.

2) “The consistency of dosing approaches included a total daily dosing of 40 to

240mg of furosemide by bolus delivery for 1 to 2 weeks, which was the most common (34–

36,39,41).” The administration of 240 mg of furosemide as a bolus is an unlikely course of action.

In my view, the discussion lacks a sufficiently analytical summary of the results presented.

6. PLOS authors have the option to publish the peer review history of their article (what does this mean?). If published, this will include your full peer review and any attached files.

Reviewer #1: No

Reviewer #2: No

---

## [Author Response · Author response to Decision Letter 0]

27 Nov 2024

Dear Reviewers,

Thank you for your review and feedback of our manuscript. We appreciate the opportunity to provide you with a resubmission to address your comments and concerns. Please see below for our comments in reference to the reviewer comments.

Reviewer #1 Comments: It would be more useful if there were more information on the results and effectiveness of IV diuretics treatment, if possible. 

Response: Thank you for your comment. Unfortunately it is difficult to fully understand the outcomes due to variability in study designs, small sample sizes, and diverse patient population. The differences in home care protocols, implementation, and patient monitoring and reporting complicate the standardization of approaches for evaluation or directly compare study results. Furthermore, it was not within the scope of this review to evaluate the outcomes and effectiveness of home IV diuretics as an intervention, but rather understand how the intervention was implemented. The authors have added a comment about this under ‘Limitations’ on page 36. 

“The major limitation of this publication is the lack of comprehensive evaluation regarding the outcomes and effectiveness of home IV diuretic therapy for HF. Fully understanding these outcomes is challenging due to variability in study designs, small sample sizes, and diverse patient population. The differences in home care protocols, implementation, and patient monitoring and reporting complicate the standardization of approaches for evaluation or directly compare study results. These limitations underscore the need for more rigorous studies and reporting structures to clarify the true impact of home IV diuretic therapy on long-term HF outcomes.”

Reviewer #2 Comments: 

The introduction is deficient in terms of global data, as it presents only the data from Canada as representative 

Response: Thank you for your comment. The introduction has been edited to reflect global data of HF burden in the Introduction on page 4.

“Heart failure (HF) poses a significant global health burden, with high rates of hospital readmission and mortality affecting patients around the world (1). Despite advances in treatment, many patients experience frequent hospitalizations and declining quality of life, highlighting the urgent need for improved HF management strategies across counties and health systems (1,2,3). The course of chronic HF is troubled by recurrent exacerbation episodes and an overall decline in health status, resulting in frequent hospitalizations and poor quality of life (3,4) Loop diuretics in the form of furosemide are the treatment of choice for symptomatic relief of HF congestion (3,5). However, as the disease progresses, patients become less responsive to oral furosemide and require intravenous (IV) administration to address issues of diuretic resistance and cardio-renal dysfunction (6,7).”

Reviewer #2 Comments: “The consistency of dosing approaches included a total daily dosing of 40 to 240mg of furosemide by bolus delivery for 1 to 2 weeks, which was the most common (34– 36,39,41).” The administration of 240 mg of furosemide as a bolus is an unlikely course of action. In my view, the discussion lacks a sufficiently analytical summary of the results presented. Thank you for your comment. Further detail has been added to the discussion about dosing approaches on page 33. 

“The dosing approaches observed in these studies included a total daily dose of 40 to 240mg of furosemide, commonly administered via bolus delivery over 1 to 2 weeks (34–36,39,41). Although specific dosing intervnals were not always explicitly outlined, standard care practices would typically suggest twice-daily dosing for total daily amounts at or above 80mg, with higher doses up to 120mg twice a day (Felker et al., 2020). Additionally, the study by Godino et al. reported a mean daily dose of 302.5 ± 176 mg; however, this dosing regimen was delivered through a continuous infusion pump rather than bolus injections, reflecting a variation in administration method that may impact overall efficacy and ability to support this intervention.”

Additional edits to the manuscript include the following:

• Change of Headings format to reflect the different levels as outlined in the formatting guideline and sentence casing. 

• The Figure title has been reformatted and referenced within the text. 

• The Table titles have been reformatted and re-ordered within the document to be included directly after the paragraph it is first cited in.

• Appendix A, which was the ‘Sample search strategy’ has been removed and resubmitted as a supplemental file, under ‘S1 File’. No alterations have been made to the content. 

We thank you for your attention to our submission and look forward to additional feedback. 

Kind regards,

Morgan Krauter, MN-NP

---

## [Decision Letter · Decision Letter 1]

17 Dec 2024

Home intravenous diuretic administration for heart failure management: A scoping review.

PONE-D-24-43583R1

Dear Dr. Krauter,

We’re pleased to inform you that your manuscript has been judged scientifically suitable for publication and will be formally accepted for publication once it meets all outstanding technical requirements.

Kind regards,

Vincenzo Lionetti, M.D., PhD

Academic Editor

PLOS ONE

Additional Editor Comments (optional):

Reviewers' comments:

Reviewer's Responses to Questions

**Comments to the Author**

1. If the authors have adequately addressed your comments raised in a previous round of review and you feel that this manuscript is now acceptable for publication, you may indicate that here to bypass the “Comments to the Author” section, enter your conflict of interest statement in the “Confidential to Editor” section, and submit your "Accept" recommendation.

Reviewer #1: All comments have been addressed

Reviewer #2: All comments have been addressed

2. Is the manuscript technically sound, and do the data support the conclusions?

Reviewer #1: Yes

Reviewer #2: Yes

3. Has the statistical analysis been performed appropriately and rigorously? 

Reviewer #1: Yes

Reviewer #2: N/A

4. Have the authors made all data underlying the findings in their manuscript fully available?

Reviewer #1: Yes

Reviewer #2: Yes

5. Is the manuscript presented in an intelligible fashion and written in standard English?

Reviewer #1: Yes

Reviewer #2: Yes

6. Review Comments to the Author

Reviewer #1: The revised manuscript was finely corrected. The manuscript has some benefits fo readers who are interested in this issue.

Reviewer #2: I have no further comments. The discussion could probably be more insightful, but it is sufficient for publication.

7. PLOS authors have the option to publish the peer review history of their article (what does this mean?). If published, this will include your full peer review and any attached files.

Reviewer #1: No

Reviewer #2: No

---

## [Editor Report · Acceptance letter]

9 Jan 2025

PONE-D-24-43583R1 

PLOS ONE

Dear Dr. Krauter, 

I'm pleased to inform you that your manuscript has been deemed suitable for publication in PLOS ONE. Congratulations! Your manuscript is now being handed over to our production team.

Kind regards, 

on behalf of

Prof. Vincenzo Lionetti 

Academic Editor

PLOS ONE